# Targeting Autophagy for Developing New Therapeutic Strategy in Intervertebral Disc Degeneration

**DOI:** 10.3390/antiox11081571

**Published:** 2022-08-14

**Authors:** Md Entaz Bahar, Jin Seok Hwang, Mahmoud Ahmed, Trang Huyen Lai, Trang Minh Pham, Omar Elashkar, Kazi-Marjahan Akter, Dong-Hee Kim, Jinsung Yang, Deok Ryong Kim

**Affiliations:** 1Department of Biochemistry and Convergence Medical Science, Institute of Health Sciences, College of Medicine, Gyeongsang National University, Jinju 52727, GyeongNam, Korea; 2College of Pharmacy and Research Institute of Pharmaceutical Sciences, Gyeongsang National University, Jinju 52828, GyeongNam, Korea; 3Department of Orthopaedic Surgery, Institute of Health Sciences, Gyeongsang National University Hospital and Gyeongsang National University College of Medicine, Jinju 52727, GyeongNam, Korea

**Keywords:** intervertebral disc degeneration, autophagy, low back pain, autophagy-targeting compounds, ECM degradation

## Abstract

Intervertebral disc degeneration (IVDD) is a prevalent cause of low back pain. IVDD is characterized by abnormal expression of extracellular matrix components such as collagen and aggrecan. In addition, it results in dysfunctional growth, senescence, and death of intervertebral cells. The biological pathways involved in the development and progression of IVDD are not fully understood. Therefore, a better understanding of the molecular mechanisms underlying IVDD could aid in the development of strategies for prevention and treatment. Autophagy is a cellular process that removes damaged proteins and dysfunctional organelles, and its dysfunction is linked to a variety of diseases, including IVDD and osteoarthritis. In this review, we describe recent research findings on the role of autophagy in IVDD pathogenesis and highlight autophagy-targeting molecules which can be exploited to treat IVDD. Many studies exhibit that autophagy protects against and postpones disc degeneration. Further research is needed to determine whether autophagy is required for cell integrity in intervertebral discs and to establish autophagy as a viable therapeutic target for IVDD.

## 1. Introduction

Low back pain is an extremely common musculoskeletal condition that is experienced by nearly everyone at some point in their lives [1,2]. As low back pain is one of the leading sources of years lived with disability worldwide, more attention is urgently needed to ease its impact on health care systems [3,4,5].

Intervertebral disc degeneration (IVDD) is a prevalent musculoskeletal disorder defined as the degeneration of one or more intervertebral discs (IVDs) that separate the vertebrae, which produces pain in the back, neck, or extremities [6]. People of all ages are affected by IVDD, which decreases their quality of life and increases clinical and economic expenditures [7,8,9]. Over 40% of cases of low back pain result from IVDD progression, which can be caused by excessive stress, aging, injury, spine deformity, and hereditary predisposition [10,11,12,13]. Mutations in IVD proteins, such as collagens, proteoglycans, cytokines, and proteolytic enzymes, and the vitamin D receptor, are linked to IVDD [14,15]. Although much research has been conducted to investigate the molecular mechanisms of IVDD, the pathogenic process underlying the disc degeneration is still not fully understood.

Autophagy is a cellular process that removes damaged proteins and malfunctioning organelles and, thereby, acts as an organismal homeostasis mechanism [16,17]. Growing evidence implicates autophagy failure in a variety of diseases, including IVDD and osteoarthritis [18,19,20,21,22]. Thus, a synthesis of the literature on the role of autophagy in IVDD is needed to chart paths toward more effective therapeutic options. In this review, we describe recent studies linking autophagy to IVDD and highlight potential treatment options based on known underlying mechanisms of the disease.

## 2. Genetics of IVDD

### 2.1. IVD Composition

IVDs support the spinal column and act as shock absorbers against the human body’s natural axial loading. The spinal column, which starts at the skull base and terminates at the coccyx, is made up of vertebrae, and IVDs consist of five regions: cervical (C1-C7), thoracic (T1-T12), lumbar (L1-L5), sacral (S1-S5), and coccygeal (Co1-Co4/5) [23]. IVDs are divided into two parts: the central nucleus pulposus (NP) and the outer annulus fibrosus (AF) area [24,25] (Figure 1). The NP is mostly made up of proteoglycans and water. It functions as a hydrogel that helps maintain hydrostatic pressure by trapping water [26]. The NP also contains collagen network components including collagen type I, collagen type II, and other materials (e.g., chondrocyte-like cells) [27]. The composition of the NP allows it to remain elastic, compressible, and flexible against stress forces. The AF mostly consists of a highly structured collagen fiber network containing collagen type I, collagen type II, and other materials (e.g., fibroblast-like cells) that efficiently restrict the NP in place [28]. Sharpey’s fibres connect the AF to the vertebral body, forming a fibrous framework around the NP [29]. The IVD also contains interdependent and structurally connected cartilaginous endplates (CEPs).

### 2.2. IVDD

IVDD most commonly affects IVDs in the lower (i.e., lumbar) region of the spine but can affect any spinal region. IVDD can produce periodic or chronic discomfort in the back or neck depending on the location of the afflicted IVDs. Pain is typically exacerbated upon sitting, bending, twisting, or lifting objects. The spine can be vulnerable to stress and aberrant gene expression due to certain biological and environmental variables. As a result, IVDs may experience increased catabolic activity and decreased anabolic activity, resulting in their degeneration. Indeed, research shows that IVDD has a complicated pathophysiology related to factors such as aging, spine abnormalities and disorders, spine injuries, and genetic factors [26,30]. However, the molecular mechanisms underlying the disease are not yet fully understood.

### 2.3. Genes Associated with IVDD Progression

In vitro and in vivo studies of IVDD reveal several genes that impact IVD extracellular matrix (ECM) anabolism and catabolism [31,32,33,34,35]. Although IVDD is a heterogeneous disorder influenced by both genetic and environmental factors, emerging data indicate that the genetic influence greatly outweighs the environmental factors. Recent research reports a six times higher risk of developing IVDD among individuals with a genetic predisposition compared with the general population [36]. Thus, IVDD has been classified as a common, oligogenic, multifactorial genetic disorder [14]. Mutations in IVD-related proteins, such as collagens, proteoglycans, cytokines, and proteolytic enzymes, and the vitamin D receptor appear to play a role in IVDD pathogenesis [15]. The genes associated with IVDD progression are depicted in Figure 2.

#### 2.3.1. Gene Products Related to the Synthesis of ECM

The ECM is a non-cellular three-dimensional macromolecular structure made up of roughly 300 proteins that are found in the extracellular environment of all tissues and participate in various cellular activities [37,38,39]. Collagens, proteoglycans, and non-collagenous proteins are the primary components of the ECM in IVDs [40,41]. Proper physiological functioning of IVDs depends on balanced ECM dynamics [7,9,42,43]. IVD cells are dispersed across a complex network of interconnected sugar molecules and collagen that communicates reciprocally with the neighboring ECM [40,44].

##### Collagens

Collagens (collagens I, II, III, IX, and XI) are crucial structural elements of the ECM. Among the twenty-eight types of collagen discovered in different parts of the IVD matrix, the AF and NP have the highest quantities of collagen I, II, III, IV, V, IV, IX, X, and XI [45]. Although some studies have investigated potential links between polymorphisms in collagen genes and risk of IVDD, they have produced inconsistent findings due to variations in participant ethnicity, age, and site of single nucleotide polymorphism genetic predisposition [15,46,47]. Table 1 lists the ECM synthesis-related collagen genes linked to IVDD pathophysiology based on the latest studies.

##### Proteoglycans

Aggrecan is a major and functionally important type of proteoglycan in the IVD extracellular matrix. Various numbers of tandem repeat (VNTR) polymorphisms within the aggrecan CS1 domain are unique to humans [61]. As a result, several recent studies have investigated the link between aggrecan VNTR polymorphism and the risk of IVDD, with conflicting findings [62,63,64,65,66,67,68,69,70,71].

##### Non-Collagenous Proteins

The IVD matrix contains a variety of non-collagenous proteins that govern pathologic pathways and induce disc dysfunction and degeneration [72,73,74,75]. 

#### 2.3.2. Proteolytic Proteins Responsible for ECM Degradation

The ECM is characterized by a dynamic architectural framework that is continually modified in both normal and pathological conditions, driven by many matrix-degrading proteolytic enzymes such as matrix metalloproteinases (MMPs), a disintegrin, and metalloprotease with thrombospondin motifs (ADAMTS), and other proteases [37].

##### MMPs

MMPs are a group of proteolytic enzymes capable of degrading key critical elements (i.e., collagens and proteoglycans) of IVDs [76]. MMP expression is low in normal IVD tissue but is increased in degenerative IVD tissue. Higher MMP expression was observed as IVDD worsens [77,78,79,80]. The elevated expression of MMP-1 and -3 in inflammatory cells in degenerating discs implies that these proteinases are linked to IVDD [81,82,83]. Table 2 lists the MMP proteolytic genes linked to the pathophysiology of IVDD based on current research findings.

##### ADAMTS

Aggrecans are cleaved at a specific “aggrecanase” site, which involves several members of the ADAMTS family [96,97,98,99,100,101,102]. Although the role of MMPs in degenerative disorders has received much attention, the most prevalent proteoglycan degradation product in IVDD contains a terminal amino acid sequence structure suggestive of aggrecanase activity [98]. Therefore, aggrecanase activity could be a key player in the progression of IVDD [103,104,105,106,107,108].

##### Other Proteases

Cathepsins are cysteine proteases of the peptidases family that break down proteins in endosomes and lysosomes; however, they are also produced in the extracellular space and are linked to ECM degradation [109,110,111,112]. The presence of cathepsins D, G, L, and K at the site of degeneration implies that these proteinases are linked to CEP separation and AF disorganization in degenerative diseases [113,114,115,116]. Other proteases linked to anabolic imbalance and pathogenesis during IVDD include high-temperature requirement A1 (HTRA1), transmembrane serine protease 1 (TMPRSS1), and heparanase isoforms (HPSE1 and HPSE2) [117,118,119,120,121].

#### 2.3.3. Other Gene Products Related to IVDD Pathogenesis

##### Inflammatory Cytokines

As inflammation contributes to IVDD, genomic alterations in inflammatory and anti-inflammatory markers may have a particular impact on IVDD. The balance of pro-inflammatory (e.g., IL-1, IL-6, and TNF-α) and anti-inflammatory (e.g., IL-10, TGF-β, and IL4) cytokines is associated with the onset and severity of IVDD [122,123]. As a result, the amount of cell death in IVDD is determined by the interplay between inflammatory and anti-inflammatory cytokines as well as other components of the IVD ECM. For example, TNF-α, IL-1, IL-2, IL-4, and IL-10 single nucleotide polymorphisms are strongly associated with IVDD in the Iranian population, suggesting that genetic modifications in anti-inflammatory cytokines contribute to IVD homeostasis imbalance and degenerative changes [123,124,125,126,127]. The expression levels of inflammatory pain mediators, prostaglandin E2 (PGE2), and cyclooxygenase 2 (COX-2) were found to be greater in degraded IVD cells [128]. In addition, human disc degeneration is linked to a new catechol-O-methyltransferase (COMT) variation [129].

##### Vitamin D Receptor

Vitamin D receptor mutations may contribute to pathologies such as osteoporosis, osteoarthritis, and IVDD because they affect bone mineralization and remodeling [130,131,132]. The vitamin D receptor gene, which is found on human chromosome 12 (12q12–q14) and has eight protein-coding and six untranslated exons, is one of the most highly investigated genes linked to IVDD [133,134,135,136,137,138,139]. In particular, the *t* allele of VDR Taq I is associated with a higher risk of developing IVDD and disc herniation, especially in people under the age of 40 [140,141].

##### Apoptotic and Growth Factors

Degenerated discs exhibit considerably higher apoptosis, according to studies on the molecular underpinnings of IVDD [142,143,144]. Even though the underlying mechanism for IVD cell death is still under investigation, mutation in some death genes has been linked to an elevated risk of IVDD, such as tumor necrosis factor alpha (TNF-α), caspase-3, capase-9, TNF-related apoptosis-inducing ligand (TRAIL), and death receptor-4 (DR-4) [145,146,147,148,149,150].

Back pain is also related to the ingrowth of blood vessels and nerves into the IVD during degeneration. Therefore, some growth factors are considered as a possible therapeutic to improve IVD tissue regeneration. Indeed, platelet-derived growth factor (PDGF), insulin-like growth factor 1 (IGF-1), growth differentiation factor 5 (GDF-5), bone morphogenetic protein 2 (BMP-2), and bone morphogenetic protein 7 (BMP-7) have been explored as viable treatment alternatives for IVD regeneration [151,152,153,154,155,156,157,158,159,160,161,162,163]. Furthermore, VEGF is highly expressed in the injured IVD, and mutational variations in the VEGF gene are associated with lumbar disc degeneration in a young Korean population [164].

##### Non-Coding RNAs (ncRNAs)

Several non-coding RNAs (ncRNAs) are dysregulated in IVDD, suggesting a role in the development of the disease [165,166,167]. Some ncRNAs’ expression profiles in IVDD patients differ from those in healthy subjects [165]. ncRNAs control apoptosis, proliferation, ECM deterioration, and inflammation during the course of the disease [167,168,169,170,171]. Other ncRNAs inhibit apoptosis and ECM degradation in normal IVD tissues through their target genes or pathways [172,173,174]. In this section, we catalog recent findings on the role of small and circular RNA in IVDD.

One study has shown that twenty-nine microRNAs (miRs) were differentially expressed in IVDD: six upregulated and twenty-three downregulated [175]. Another study has found twenty-five elevated and twenty-six miRNAs in IVDD patients [176]. miR-27a [177], miR-494 [178,179], miR-30d [180], miR-222-3p [181], miR-15a [182], miR-143 [183], miR-532 [184], miR-138-5p [185], miR-106a-5p [186], miR-34a [187], and miR-221 [188] have been suggested as potentially pro-apoptotic in IVDD. On the other hand, miR-155 [175], miR-21 [189], miR-499a-5p [190], miR-486-5p [191], miR-125a [192], miR-145 [193], and miR-573 [194] were significantly downregulated in IVDD tissues. This subsequently suppressed apoptosis through the interaction with mRNA of PTEN, SOX4, FOXO1, TP53INP1, ADAM17, and Bax, respectively. Moreover, miR-16 [195], miR-194 [196], and miR-223 [197] exhibit a protective effect against LPS-induced inflammation and IVDD. 

Studies have demonstrated the role of circular RNAs (circRNAs) in IVDD. HDC-Circ 0083756 promotes IVDD by inhibiting the miR-558/TREM1 axis, representing a therapeutic target for IVDD [198]. An abnormally high amount of miR-200c promotes apoptosis of NP cells and ECM degradation by suppressing XIAP. CircVMA21 attenuates the activity of miR-200c, but it is highly suppressed in IVDD tissues [199]. Similarly, Circ-GRB10 is downregulated in degenerative NP cells. Overexpressing GRB10 reduces apoptosis of NP cells via miR-328-5p and activation of the ErbB pathway [200]. In addition, Circ-4099 induces collagen II and aggrecan production and inversely reduces the production of pro-inflammatory molecules, including IL-1, TNF-, and PGE2 via miR-616-5p inhibition [201].

In summary, individual genes and external factors may interact in the etiology of IVDD, producing gene-to-gene, gene-to-environment, and gene-to-age interactions [47]. More research, including linkage analysis and whole-genome scans of people of diverse ethnicities across the lifespan, is needed to improve our understanding of the genetic influence on IVDD and to uncover additional genes involved in IVDD pathogenesis.

## 3. Current Treatment Approaches for IVDD

There is an increase in the prevalence of intervertebral disc diseases in both young and older populations [202,203,204]. Effective therapeutic approaches are needed. There is a substantial body of research on the treatment of IVDD, which we can divide into three major categories: (a) conservative or non-surgical management, (b) surgical treatment, and (c) molecular and biological therapy [204,205,206].

(a)Conservative or non-surgical management

Some IVDD patients benefit from conservative care, which includes physiotherapy, oral analgesics, vitamins, exercise, heat, cold, corsetry, acupuncture, radio frequency/shock waves, and varying intensities of massage [204,205,207]. However, the condition worsens for some patients or frequently recurs [208]. This approach is generally safe but with modest outcomes.

(b)Surgical treatment

Surgery may include decompression to relieve neurological symptoms [209,210], fusion to stop motion at a functional spinal unit [211,212], motion preservation/modifying surgery using disc replacement/dynamic fixation devices [213,214], and deformity surgery to correct biomechanics across several functional spinal units [215,216]. Patients who undergo surgery have poorer baseline conditions but recover and return to normal quicker, even though the long-term outcomes may vary [210]. Nevertheless, many people have post-spinal surgery syndrome (PSSS), which is chronic or recurring LBP following surgery [217,218,219]. In addition, the effectiveness of these surgical procedures is not always adequate and may result in unfavorable side effects, including a recurrence of nucleus pulposus (NP) herniation [220,221,222], neighboring segment infection [223], or surgical site infection [224]. Therefore, there is a need for more direct and efficient therapies.

(c)Molecular and biological therapies

Molecular therapies hold great promise for the treatment of IVDD. Compared to the conservative approach, they do not focus on pain relief or invasive surgeries that replace or fuse the affected disc. Several experimental and clinical trials have been devised to regenerate and repair the damaged disc. These include manipulating different growth factors with or without carriers [158,225,226,227,228], cells with or without scaffolds [32,229,230,231,232], and gene therapy [233,234]. Effective gene transfer to target cells inside IVD in animal models offers promising new opportunities for IVDD therapy, for example, gene transfer to disc cells through targeting RNA interference (RNAi), clustered regularly interspaced short palindromic repeats (CRISPR), and mammalian target of rapamycin (mTOR) signaling [234,235]. Currently, a limited number of efficient biological agents for IVD regeneration have only been tested in animal models [236,237,238,239]. The role of autophagy in the development of IVDD has been rigorously studied. Autophagy may be a double-edged sword in developing IVDD [240,241,242,243,244,245]. Modulating autophagy represents a potential target for IVDD therapy.

## 4. Autophagy as a Potential Therapeutic Target for IVDD

### 4.1. Autophagy and Its Implications for IVDD

Autophagy is a self-digesting system that allows cells to sequester internal and external substrates in double-membrane structures known as autophagosomes. Autophagosomes merge with lysosomes to degrade their contents. Autophagy maintains cellular homeostasis and contributes biosynthesis and energy generation in the face of nutritional and metabolic stresses [246]. Several proteins are essential for the localization of autophagy-related gene (ATG) proteins to the autophagosome formation site, called the pre-autophagosomal structure [247]. A complex containing unc-51-like kinase 1 (ULK1) and members of the ATG family, which includes ATG13, focal adhesion kinase family interacting protein of 200 kDa (FIP200), and ATG101, initiates the autophagic process [248,249]. The ULK1 complex recruits the vacuolar protein sorting 34 (VPS34) complex, which contains VPS34, BECN1, VPS15, and ATG14L at autophagy initiation sites [248,250]. VPS34 complex proteins help form the phagophore that is then processed into an autophagosome through lipidation of LC3 with phosphatidylethanolamine (PE) by another complex that includes ATG16, ATG5, and ATG12 [251]. Autophagy makes an essential contribution to the development of IVDD, with most recent studies indicating that abnormal autophagy levels and aberrant nutrition in IVDs are important factors leading to IVDD [252,253,254,255]. However, the role of autophagy in the IVDD process is still debated. Although most recent studies show that autophagy protects against and delays IVDD, some studies show that autophagy can accelerate IVDD.

Several studies show that autophagic flux, which encompasses the complete dynamic process of autophagy, is linked to the development of IVDD (Figure 3). For example, restoring the autophagic flux protects against IVDD by reversing oxidative damage and mitochondrial dysfunction [256]. Thus, activating autophagy can be a potential target for treating IVDD.

### 4.2. Autophagy-Targeting Therapeutic Approaches for IVDD

Autophagy is a regulatory mechanism whereby autophagosomes containing organelles or cytosolic proteins are ultimately destroyed by lysosomes to preserve normal physiological equilibrium and stability [257,258]. Autophagy and apoptosis of NP cells in response to oxidative stress are linked to IVDD [241,259]. Despite the existence of many studies investigating autophagy and IVDD, therapeutic approaches that target autophagy are still limited. In this section, we discuss molecules and compounds that have been used to target autophagy for the treatment of IVDD.

#### 4.2.1. Autophagy-Targeting ncRNAs in IVDD

ncRNA is a type of RNA that is transcribed from DNA but does not have the ability to be translated into proteins or peptides. ncRNA includes short hairpin RNA (shRNA), small interfering RNA (siRNA), antisense RNA, microRNA (miRNA), long non-coding RNA (lncRNA), circular RNA (circRNA), and extracellular RNA [260,261,262]. Although evidence suggests that miRNA, lncRNA, and circRNA contribute to the progression of IVDD by affecting the processes of apoptosis, cell proliferation, ECM degradation, and inflammation [263,264,265,266], the role of ncRNA in autophagy is unclear. Figure 4 depicts autophagy-targeting ncRNAs that can be exploited to treat IVDD.

##### miRNA

miRNAs, which consist of a unique family of 19–25-nucleotide microscopic ncRNAs, are hypothesized to govern several biological activities including cell development and death, cellular senescence, and inflammatory cytokine release [267,268,269,270,271]. Growing evidence indicates that miRNAs play a major role in the genesis and progression of malignancies by acting as tumor suppressor genes or oncogenes [272,273,274,275,276]. Recent research also reports that miRNAs are related to IVDD. For example, one study reports that miR-184 expression is elevated in degenerative NP cells and is positively associated with IVDD severity [277]. By contrast, another study shows that miR-573 expression is downregulated in degenerative NP cells, along with reduced Bcl-2 expression and elevated Bax, cleaved caspase-9, and cleaved caspase-3 expression [194]. The results of several recent studies suggest that miRNA-regulated autophagy can be a potential therapeutic target in IVDD. In addition, the increased expression of miR-21 suppresses autophagy in many cell types including chondrocytes [278,279,280]. In human degenerated NP cells, miR-21 specifically induces ECM breakdown, which markedly downregulates PTEN expression and consequently results in Akt activation [174]. By targeting SIRT1 via PTEN/PI3K/Akt signaling, miR-138-5p enhances TNF-induced apoptosis in human IVDD [185]. miR-10b targets HOXD10 and stimulates NP cell proliferation via the RhoC-Akt pathway in IVDD [281]. Furthermore, activating the Akt/mTOR pathway inhibits autophagy, which enhances MMP-3 and -9 production and facilitates Col II and aggrecan breakdown [282]. Notably, miR-129-5p inhibits autophagy by targeting beclin-1 in NP cells. Indeed, the autophagy activity decreases in human NP cells transfected with a miR-129-5P mimic. By contrast, its activity increases upon treatment of a miR-129-5P inhibitor via modulation of Beclin-1 expression [283]. Moreover, overexpression of miR-185 exhibits a decrease in the autophagy activity by suppressing the Wnt/β-catenin signaling pathway via galectin-3 [284]. Similarly, miR-210 suppresses autophagy in human degenerated NP cells by directly targeting ATG7 and upregulating MMP-3 and -13 expression, leading to enhanced degradation of Col II and aggrecan [285]. Another functional assay suggests that miR-654-5p inhibits autophagy and facilitates ECM degradation by boosting MMP-3, -9, and -13 expression and lowering collagen I, collagen II, SOX9, and aggrecan expression via increased levels of phosphorylated (p)-PI3K, p-AKT, and p-mTOR [286]. Furthermore, expression of MMP-14 is induced by miR-193a-3p downregulation, which accelerates the loss of type II collagen and, hence, promotes IVDD [287]. Notably, overexpression of miR-15a promotes NP cell proliferation and triggers apoptosis by downregulating MAP3K9, upregulating Bax and caspase-3, and downregulating Bcl-2 in degenerative NP tissue and cells [182]. In addition, miR-96 stimulates the growth of human degenerated NP cells by targeting ARID2 via the Akt pathway, suggesting that it could serve as a therapeutic target for IVDD [288].

##### lncRNA

lncRNA has a limited ability to code for proteins and is involved in a variety of biological processes including transcription, protein activity, and aging-related degenerative musculoskeletal disorders such as IVDD [289]. In IVDD, dysregulated lncRNA plays a key role in altering NP cell activity [290]. Increasing evidence suggests that lncRNA interacts with miRNA, inducing cell autophagy and death through a lncRNA–miRNA–mRNA competitive endogenous RNA (ceRNA) network [167,291]. The lncRNA-FAM83H-AS1 pathway preserves IVD tissue homeostasis and reduces inflammation-related discomfort by inhibiting miR-22-3p, which promotes NP cell proliferation [292]. lncRNA-H19 increases autophagy and death in NP cells, exacerbating IVDD via the miR-139/CXCR4/NF-B axis [293]. In addition, the lncRNA HOTAIR induces apoptosis, senescence, and ECM catabolism in NP cells by upregulating autophagy [294]. HOTAIR is highly abundant in degenerative NP cells, and si-HOTAIR reduces apoptosis and autophagy in degenerative NP cells by boosting PTEN expression as a ceRNA of miR-148a [295]. HOTAIR regulates degenerative changes in IVDs via the Wnt/β-catenin pathway [296] and impacts cell proliferation in IVDD via the miR-130b/PTEN/AKT axis [297]. Furthermore, under nutrition deficiency stress, long intergenic non-protein coding RNA 641 (LINC00641) regulates autophagy and the development of IVDD by serving as a ceRNA of miR-153-3p [298].

##### circRNA

circRNA regulates gene transcription and translation as well as inflammatory cytokine release, ECM metabolism, and cell proliferation and death [266], suggesting that it could be therapeutically targeted in IVDD. However, our understanding of the mechanism by which circRNA impacts IVDD is still limited. circRNAs were recently reported to influence pathogenic processes of IVDD including inflammation, ECM metabolism, NP cell proliferation, autophagy, and apoptosis by acting as ceRNAs [266,299,300,301]. Thus, these newly discovered ceRNA interaction sites could be important targets for IVDD treatment [167,291]. A growing body of evidence suggests that circRNA and miRNA control cell death and autophagy in IVDD. For example, through the miR-299-5p/Galectin-3 axis, circRNA-RERE promotes oxidative stress-induced apoptosis and autophagy in NP cells [302]. Similarly, circRNA-ERCC2 alleviates IVDD by regulating autophagy and apoptosis via the miR-182-5p/SIRT1 axis [303]. Moreover, circRNA-CIDN binds to miR-34a-5p and prevents compression loading-induced NP cell injury by targeting SIRT1, suggesting that it could be therapeutically targeted to treat IVDD [304].

#### 4.2.2. Autophagy-Targeting Compounds in IVDD

Although autophagy may have a dual role in IVDD, most recent investigations show that it protects against IVDD. Autophagy reduces NP cell apoptosis, ECM degradation, senescence, and CEP inflammation and calcification, thereby balancing levels of IVD ECM components and preventing IVDD [22,254,285,305,306,307,308,309]. A growing number of studies suggest that targeting autophagy could be a promising therapeutic strategy for treating IVDD. A variety of compounds have been suggested to slow IVDD progression by regulating autophagy activity in vivo and in vitro through different signaling pathways and functional proteins (Figure 5). 

##### Alleviating Apoptosis, Cell Senescence, and ECM Degradation in IVDD

Inhibiting apoptosis is a promising strategy for treating IVDD. The exogenous death receptor pathway, endogenous mitochondrial pathway, and endoplasmic reticulum stress pathway are among the apoptotic pathways linked to IVDD [310,311]. These pathways may be effective therapeutic targets because they cause IVDD by inducing IVD cell death, which is the most prevalent apoptotic component in IVDD [312]. For example, puerarin, an 8-C-glucoside of daidzein extracted from Pueraria plants, prevents apoptosis and cell death in in vitro human NP mesenchymal stem cells (NPMSCs) and in a rat compression model by maintaining mitochondrial membrane potential and reducing reactive oxygen species generation by activating the PI3K/Akt pathway [313]. Additionally, cyclosporine reduces apoptosis in NPMSCs by reducing mitochondrial malfunction and oxidative stress [314]. In rat NPMSCs, naringin, a bio-flavonoid extracted from tomatoes, grapefruit, and related citrus fruits, inhibits hydrogen peroxide (H_2_O_2_)-induced apoptosis via the PI3K/Akt pathway. Autophagy induced by oxidative stress induces apoptosis of NP cells, and autophagy mediated by mechanical stress induces apoptosis of AF cells [241]. On the other hand, autophagy triggered by hypoxia or metformin protects NP cells from apoptosis [242,244]. 

As an intricate interplay between autophagy and apoptosis maintains healthy IVDs, apoptosis signaling pathways or proteins targeting autophagy could be exploited to treat IVDD. For example, upregulation of transcription factor EB (TFEB) increases autophagy of NP cells, reverses TBHP-induced autophagic flux degradation, and decreases cleaved caspase-3 expression [22]. Similarly, downregulation of STING reduces apoptosis, senescence, and ECM degradation [315]. Berberine inhibits apoptosis and ECM breakdown in NP cells in a mouse model of disc degeneration [316]. Cyanidin inhibits rat NP cell apoptosis and IVDD via modulation of autophagy and the JAK2/STAT3 signaling pathway in vitro and in vivo [317]. Sinomenine, an alkaloid monomer extracted from the Sinomenium acutum, reduces IVDD by inhibiting apoptosis and autophagy in vitro and in vivo [318]. Ecdysterone protects NP cells from apoptosis and attenuates IVDD by stimulating autophagy [319]. Furthermore, moxibustion, a traditional Chinese medical intervention, enhances autophagy and reduces the apoptosis of NP cells via the HIF-1α/VEGF pathway [320].

In addition to increased apoptosis that diminishes IVD cells and ECM, IVDD is also associated with increased cellular senescence [321,322,323]. Cellular senescence in IVDD is linked to decreased cell proliferation, hampered self-repair, an increased inflammatory response, and increased catabolic metabolism [324,325,326]. Autophagy, a well-known and widely conserved intracellular degradation mechanism, protects cells by removing senescent organelles and misfolded proteins [327]. Several studies show that autophagic flux is linked to cell senescence and apoptosis in IVDD. For example, by restoring autophagic flux, TFEB protects NP cells from apoptosis and senescence [22]. An in vitro and in vivo study suggests that BRD4 inhibition suppresses the senescence and apoptosis of NP cells by inducing autophagy in IVDD [328]. SIRT3 helps prevent IVDD by delaying oxidative stress-induced NP cell senescence [329]. Apigenin is a flavonoid that inhibits tert-butyl hydroperoxide (TBHP)-induced apoptosis, senescence, and ECM degradation by restoring autophagic flux in vitro and slows the progression of IVDD in rats in vivo [330]. The mTOR pathway activated by parathyroid hormone 1–34 reduces senescence in rat NP cells by inducing autophagy [331]. Moreover, metformin protects NP cells from apoptosis and senescence by stimulating autophagy and alleviating IVDD in vivo [242]. 

Damage and limited ability to replenish the ECM in the NP negatively impact the biomechanical properties of IVDs, which compromises spinal stability during disc degeneration [7,9,332,333]. Thus, therapeutic targets for maintaining ECM equilibrium may be crucial for preventing disc degeneration. For example, hydroxysafflor yellow A inhibits TBHP-induced oxidative stress in an NP cell line and modulates ECM equilibrium through carbonic anhydrase 12 (CA XII) [334]. Under oxidative stress, resveratrol promotes ECM production in NP cells by activating autophagy via the PI3K/Akt pathway [335]. ECM degradation and apoptosis are suppressed by eicosapentaenoic acid-induced autophagy, which slows the development of IVDD [336]. Melatonin inhibits the development of IVDD in vivo and in vitro by promoting autophagy via the NF-B signaling pathway, which prevents ECM degradation in IVDs [337]. Berberine inhibits the NF-kB pathway, which protects human NP cells from IL1-induced ECM breakdown and death [338]. Simvastatin inhibits IL-1-induced apoptosis and ECM degradation in NP cells by suppressing the NF-kB and MAPK pathways [339]. Through modulation of p38 MAPK-mediated autophagy, quercetin improves ECM integrity in IVDD patients by raising collagen II and aggrecan levels and reducing MMP13 levels [340]. In a rat model of IVDD, Duhuo Jisheng decoction inhibits ECM degradation and death in human NP cells and alleviates disc degeneration [341]. Spermidine controls the expression of anabolic and catabolic proteins involved in ECM synthesis and breakdown [342]. In NP cells, metformin boosts the expression of anabolic genes such as Col2a1 and Acan and reduces the expression of catabolic genes such as Mmp3 and Adamts5 [242]. The antioxidant and anti-inflammatory agent 6-gingerol reduces reactive oxygen species levels, inhibits NPMSC apoptosis mediated by H_2_O_2_, and protects the ECM from degradation [343]. 

##### Preventing Inflammation and Endplate Chondrocyte Calcification

During the degeneration of IVDs, secreted inflammatory mediators such as TNF-α, IL-1, IL-6, IL-8, IL-2, IL-17, IL-10, IL-4, IFN-α, and PGE2 promote ECM degradation and IVD cell autophagy, senescence, and apoptosis, ultimately resulting in low back pain [143,308,344,345]. Therefore, these mediators could be exploited to treat IVDD. Acacetin reduces inflammation and ECM degradation in NP cells and attenuates IVDD in vitro [346]. Ligustilide protects NP cells from IL-1-induced inflammation, apoptosis, and ECM degradation and slows IVDD in vivo [347]. In addition, glucosamine reduces IL-1β-induced inflammation in IVDD by triggering autophagy via the mTOR-dependent pathway [348]. 

IVDs obtain nutrients mainly through diffusion via the calcification of endplate chondrocytes, which inhibits nutrient utilization and metabolite exchange and promotes IVDD [349,350,351]. Therefore, preventing the calcification of endplate chondrocytes is critical. CEP stem cells (CESCs) may increase NP cell proliferation via a paracrine mechanism mediated in part by the SDF-1/CXCR4 axis via the ERK1/2 signaling pathway [352]. Chlorogenic acid, through inhibiting NF-κB signaling, slows CEP degeneration and alleviates IVDD [353]. Furthermore, autophagy acts as a protective response against oxidative damage to endplate chondrocytes [354]. For example, autophagy protects endplate chondrocytes against calcification caused by cyclic mechanical stress [309]. Exosomes released by CESCs to NP cells stimulate Akt/autophagy, which inhibits IVDD [355]. In addition, curcumin promotes autophagy, which reduces the deterioration of CEP caused by stress [356].

#### 4.2.3. Suppressing Autophagy as a Therapeutic Option for IVDD

Several studies have linked autophagy with the initiation of disk degeneration and the development of IVDD [240,245,252,357,358]. The NP and AF cells isolated from non-degenerative adult rats exhibit a basal level of autophagy activity under normal physiological conditions. Therefore, autophagy is required for preserving normal disk cell integrity and survival [359,360]. However, other studies of degenerative rat NP and AF cells have shown heightened autophagy activity and upregulated autophagy-related genes, such as Beclin-1, LC3, Atg12, presenilin 1, and cathepsin B, in comparison to healthy AF tissues [252]. Likewise, IVDD progresses with age, possibly through activation of macroautophagy and chaperone-mediated autophagy (CMA) [357,358]. These findings show that the interaction between autophagy and apoptosis or ROS-dependent endoplasmic reticulum stress could cause IVDD [245].

Blocking autophagy can lessen IVDD. For example, autophagy induced by oxidative or mechanical stress triggers apoptosis in NP and AF cells [41,245]. Indeed, in a rat model of menopause, estradiol reduces IVDD by modulating antioxidant enzymes and inhibiting autophagy [361]. In addition, TGF-b1 rescues AF cells from starvation-induced apoptosis by suppressing excessive autophagy through PI3K-AKT-mTOR and MAPK-ERK1/2 [362]. Furthermore, TGF-b1 inhibits autophagy and apoptosis evoked by exogenous H2O2 in AF cells by downregulating ERK expression through the upregulation of GPx-1 [363].

## 5. Challenges and Future Applications of Autophagy-Based Therapies for IVDD

Autophagy appears to play a critical role in degenerative IVD cells, although the direction of this role varies by condition. Therefore, future research should focus on determining the specific functional aspects of autophagy in disk degeneration. Furthermore, autophagy-targeted natural molecules and/or gene therapy might be an alternative to conventional IVDD therapy. We suggest that pharmacologically targeting autophagy or gene therapy for autophagy-related genes such as mTOR, LC3, Beclin-1, p62, or other ATG genes are potential therapeutic options for degenerative disc diseases. Since autophagy is required for homeostasis under normal conditions and adaptive pathological responses, the best autophagy-targeted treatment approach for IVDD should attempt to normalize the levels of autophagy inside the IVD. 

Although RNAi-mediated mTOR signal inhibition protects human disc NP cells from degeneration, pharmacological modulators are preferable to gene-silencing treatment in clinical settings [364,365]. Indeed, the mTORC1 inhibitor rapamycin and its derivatives, such as everolimus and temsirolimus, increase mammalian lifespan and protect chondrocytes [366,367]. Recently, we reported that delphinidin protects chondrocytes against age-related oxidative stress through autophagy, with potential application in treating osteoarthritis [368]. Thus, we envision extending the use of natural compounds such as delphinidin as autophagy-based therapy for IVDD. In order to properly define the efficacy and safety of autophagy-targeted treatment, preclinical animal and clinical studies are urgently needed.

## 6. Conclusions

Accumulating evidence indicates that autophagy plays a critical role in the progression of IVDD and, thus, could be used as a novel therapeutic target. This review of recent studies on autophagy-regulated IVD degeneration lays a foundation for further research aimed at elucidating the underlying mechanisms of IVDD and understanding the contribution of autophagy, the ultimate goal of which is to develop novel approaches to IVDD treatment and prevention.

## Figures and Tables

**Figure 1 antioxidants-11-01571-f001:**
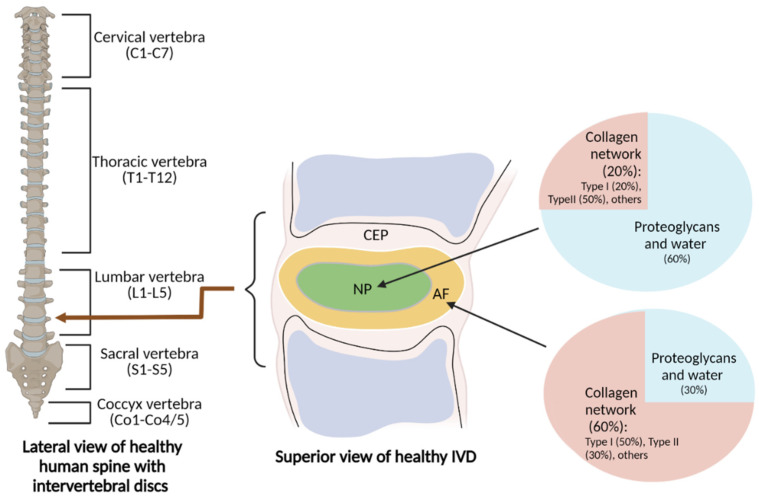
The major components of intervertebral discs. Intervertebral discs (IVDs) are located across the cervical, thoracic, lumbar, sacral, and coccygeal regions of the spinal column. IVDs consist of a central NP (nucleus pulposus), outer AF (annulus fibrosus) area, and interdependent and structurally connected CEP (cartilaginous endplate). The region-specific ECM structures, which include the proteoglycan-rich NP and collagen-rich AF, are maintained by separate cell populations.

**Figure 2 antioxidants-11-01571-f002:**
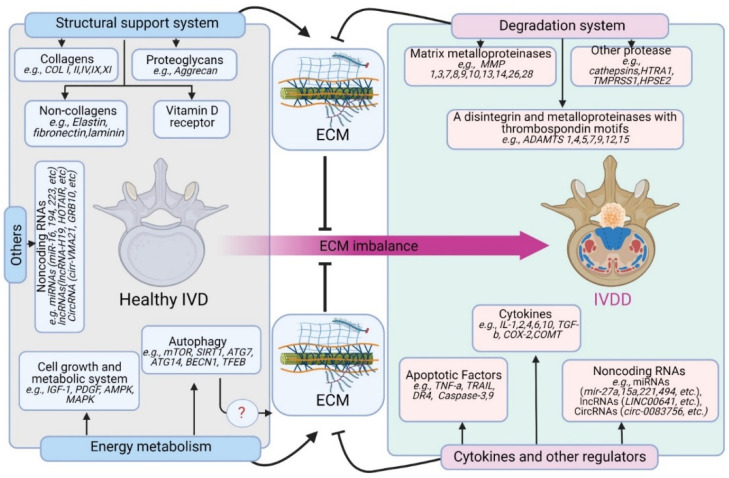
IVDD-associated genes. Mutations in IVD genes, such as structural proteins (e.g., collagens, proteoglycans, and others) and degradation enzymes (e.g., MMPs, ADAMTS, and others) are involved in the pathophysiology of IVDD associating with extracellular matrix (ECM) homeostasis. Many other gene products such as cytokines, apoptotic factors, as well as noncoding RNAs are associated with IVDD progression. Alterations in autophagy and energy metabolism genes facilitate the progression of IVDD, although their underlying mechanisms are not clear.

**Figure 3 antioxidants-11-01571-f003:**
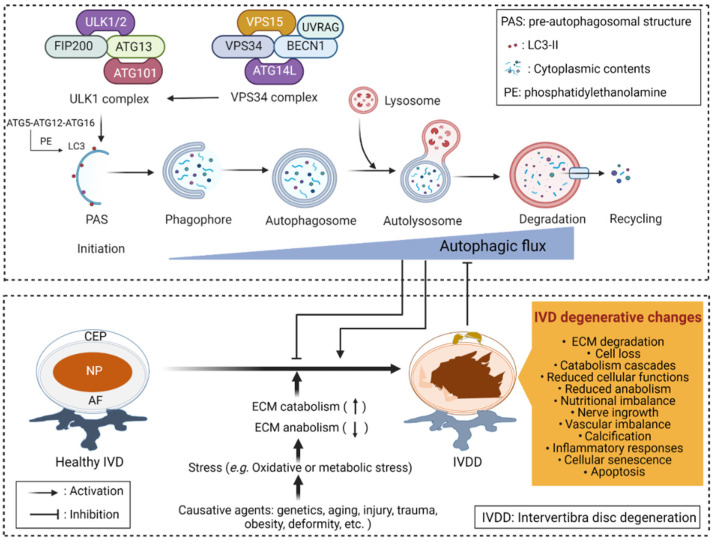
Autophagy and IVDD. Autophagic flux refers to the entire process of autophagy, including autophagosome formation, maturation, fusion with lysosomes, subsequent breakdown, and macromolecule release back into the cytosol. Excessive stress, aging, injury, spine deformity, genetic predisposition, and other pathogenic factors contribute to IVDD progression. IVDD is associated with structural and morphologic abnormalities, nutritional imbalances, nerve and vascular in-growth, decreased cellular anabolic function, activating catabolic cascades, elevated chondrocyte calcification, increased cell senescence and apoptosis, and altered expression of ECM and inflammatory markers. Autophagic flux may either delay or accelerate the onset of IVDD.

**Figure 4 antioxidants-11-01571-f004:**
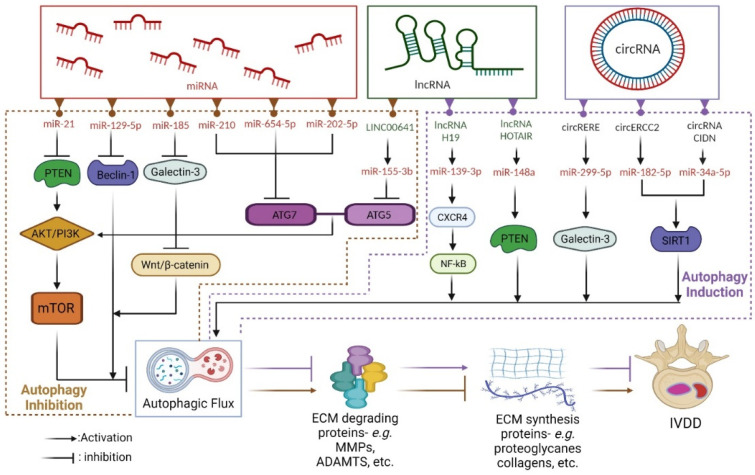
Autophagy-targeting ncRNAs for the treatment of IVDD. ncRNAs including miRNA, lncRNA, and circRNA could play important roles in IVDD treatment. These molecules can either activate (purple dotted box) or inhibit (brown dotted box) the autophagic flux, resulting in less degradation of ECM synthesis genes in IVDD.

**Figure 5 antioxidants-11-01571-f005:**
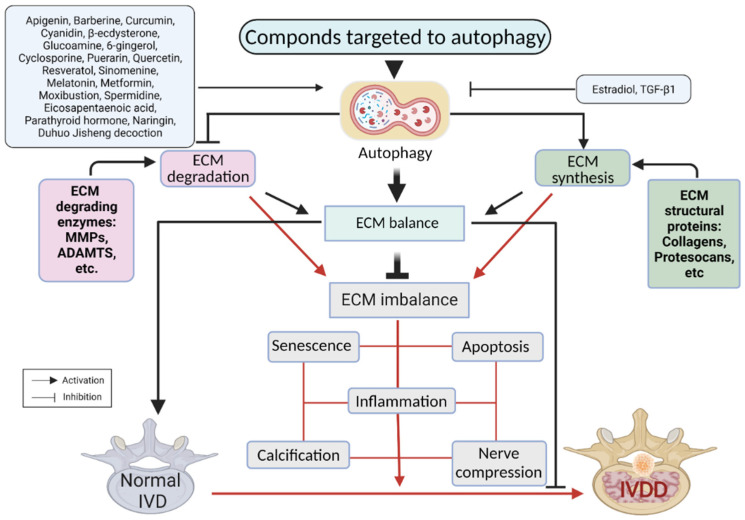
Autophagy-targeting compounds for the treatment of IVDD. Activating compounds (i.e., apigenin) targeting autophagy could reduce IVD cell apoptosis, ECM breakdown, senescence, and CEP inflammation and calcification, thereby balancing levels of IVD ECM components and preventing IVDD. Autophagy inhibiting compounds (i.e., estradiol) reverse this reaction. Black and red lines represent reactions at the balanced or imbalanced ECM condition, respectively.

**Table 1 antioxidants-11-01571-t001:** ECM synthesis-related collagen genes associated with IVDD.

Protein	Encoded Genes	Remarks	Ref.
Collagen I α-1	*COL1A1*	The rs1800012 polymorphism in COL1A1 is linked to IVDD susceptibility.	[48,49]
Collagen I α-2	*COL1A2*	COL1A2 regulates collagen activity during development and mature tissue regeneration.	[50,51]
Collagen II α-1	*COL2A1*	COL2A1 polymorphisms are linked to IVDD risk and clinical pathological characteristics in a Chinese Han population.	[52]
Collagen IX α-1	COL9A1	COL9A1 is strongly cross-linked to type II collagen and other type IX collagen molecules in NP cells.	[53]
Collagen IX α-2	COL9A2	IVDD is linked to collagen IX α chains encoded by different alleles.	[54,55,56]
Collagen IX α-3	COL9A3	IL-1β gene mutation alters COL9A3 gene polymorphism, which modifies the effect of obesity on IVDD.	[57,58]
Collagen XI α-1	*COL11A1*	A COL11A1 genetic variation is functionally linked to IVDD in a Chinese population.	[59]
Collagen XI α-2	*COL11A2*	COL11A2 polymorphism is linked to IVDD in a Chinese Han population.	[60]

IVDD: intravertebral disc degeneration; ECM: extracellular matrix; NP: nucleus pulposus; IL-1β: interleukin 1 beta.

**Table 2 antioxidants-11-01571-t002:** Proteolytic MMP genes associated with ECM degradation in IVDD.

Protein	Alternative Name	Encoded Genes	Remarks	Ref
MMP-1	Interstitial collagenase, fibroblast collagenase	*MMP1*	MMP-1 IHC score predicts the degree of disc degeneration, and inhibition of the matrix-degrading function of MMP-1 attenuates early-stage IVDD.	[84,85]
MMP-2	Gelatinase A	*MMP2*	IVD AF cells use MMP-2 to facilitate local collagen breakdown and remodeling.	[86]
MMP-3	Stromelysin-1	*MMP3*	Controlling the TNF-dependent production of MMP-3 helps halt IVDD and ECM catabolism.	[87]
MMP-7	Matrilysin	*MMP7*	Larger numbers of MMP-7 immuno-positive NP cells are indicative of intermediate and severe degrees of IVDD.	[88]
MMP-8	Neutrophil collagenase, PMNL collagenase	*MMP8*	MMP-8 is moderately upregulated in IVDD.	[80]
MMP-9	Gelatinase B, progelatinase B	*MMP9*	MMP-9 is released in the early stages of secondary damage during IVDD, and its presence in the CSF is indicative of serious IVDD and poor prognosis.	[89,90]
MMP-10	Transin-2	*MMP10*	Increased MMP-10, nerve growth factor, and substance P expression are associated with painful IVDD.	[91]
MMP-13	Collagenase 3	*MMP13*	IL-17 may activate NF-kB signaling and lead to upregulated MMP-13 expression and ECM degradation in IVDD.	[92]
MMP-14	Membrane-type MMP	*MMP14*	MMP-2 activity is linked to MMP-14 levels in IVDD.	[93]
MMP-26	Matrilysin-2, endometase	*MMP26*	MMP-26 is constitutively expressed in human IVDs in vitro and in vivo.	[94]
MMP-28	Epilysin	*MMP28*	MMP-28 exists in the ECM of more deteriorated discs and is constitutively produced in human IVD tissue.	[95]

IVD: intravertebral disc; IVDD: intravertebral disc degeneration; ECM: extracellular matrix; IHC: immunohistochemistry, AF: annulus fibrosus; NP: nucleus pulposus; CSF: cerebrospinal fluid; IL-17: interleukin 17.

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
