# Peer review of "Targeting Autophagy for Developing New Therapeutic Strategy in Intervertebral Disc Degeneration"

_antioxidants, 2022, doi:10.3390/antiox11081571_

Round 1
Reviewer 1 Report
Autophagy is an important process for IVD homeostasis. Many studies have revealed the protection role of autophagy in the progression of IDD. Kim et al had a general review on the role of autophagy in IDD. Generally, this is a well manipulated review with many understandable pictures. However, some improvements are necessary before suitable for acceptance.
1. Fig2 indicates the negative effects of some ncRNAs on the ECM imbalance during IDD. However, many ncRNAs were reported to have positive role of IDD prevention. Please discuss about this.
2. There are already many reviews introduced the function of catabolism and anabolism factors in IDD progression. This review spent almost a half to introduce these factors, making it lengthy. Please simplify it.
3. This review mainly introduced the positive role of autophagy in IDD prevention. However, some studies and reviews indicated that it is a double-edged sword in IDD. Please discuss more and add more references about this.
4. Authors indicated the potential of autophagy related therapy for IDD. Please specify the details of your expectations and new breakthrough of autophagy based—therapy (drug design).
5. Autophagic death may also be harmful to IVD. How to prevent overwhelmed autophagy in the future application of autophagy based therapy on IDD? Please give your expectations.
Author Response
"Please see the attachment."

Reviewer 2 Report
IVVD or Intervertebral Disc Degeneration is the prevalent cause of back pain and possible reason for most of the Spine related troubles. In the day-to-day daily life of a common person, he/she may probably come across this extremely common musculoskeletal condition. Elimination of IVVD very essential to ensure the quality of living. This article aims to explain how autophagy can be effectively used in the treatment of IVVD.
The terms discussed in this article are Genetics of IVVD and how Autophagy acts as a potential therapeutic target for IVDD.
From the view point of a reviewer, according to me this article is significantly contributing to the field in the research of treatment of IVVD. Article is organized and the language used is good and understandable.
Some addition small section in the end should be added and small correction is required:
As IVVD is one of the most common diseases we see in our society, while describing a therapeutic approach to this disease I suggest current and present methods used in the treatment of IVDD should be emphasized in the article. The limitations of present therapeutic techniques and how Autophagy is different and better from the present techniques can increase the significance of this article. The difficulties faced by the patients suffering from IVDD and how autophagy acting as a therapeutic agent can solve these problems should be manifested. This article is mostly based on the biochemistry point of view. It should also be on the Neurosurgical and Medicinal point of view which may help on the research attributes.
As a reviewer I suggest some minor revision and request to consider the above mentioned which may help on the future research of using autophagy as a potential therapeutic target for IVDD.
After small corrections the paper can be published
Author Response
"Please see the attachment."
